# An odd thermodynamic limit for the Loschmidt echo

Gianpaolo Torre,[1] Vanja Marić,[2,3] Domagoj Kuić,[2] Fabio Franchini,[2] and Salvatore Marco Giampaolo[2]

[1]*Department of Physics, Faculty of Science, University of Zagreb, Bijenička cesta 32, 10000 Zagreb, Croatia.*
[2]*Ruđer Bošković Institute, Bijenička cesta 54, 10000 Zagreb, Croatia*
[3]*SISSA and INFN, via Bonomea 265, 34136 Trieste, Italy*
(Dated: July 7, 2021)

Is it possible to immediately distinguish a system made by an Avogadro's number of identical elements and one with a single additional one? In this work, we show that a simple experiment can do so, yielding two qualitatively and quantitatively different outcomes depending on whether the system includes an even or an odd number of elements. We consider a typical (local) quantum-quench setup and calculate a generating function of the work done, namely, the Loschmidt echo, showing that it displays different features depending on the presence or absence of topological frustration. We employ the prototypical quantum Ising chain to illustrate this phenomenology, which we argue being generic for antiferromagnetic spin chains.

When the number of elements that make up a system is small, it is hardly surprising that even intensive quantities can be extremely sensitive to the increment or reduction of such a number. But as the amount of components increases, the common thought is that the effect of adding/removing one of them becomes less and less relevant. At the end of the process, the system reaches a point where all the intensive quantities become independent of the total number of components while the extensive ones are characterized by a simple linear dependence. In practice, while, by observing an intensive quantity, it is possible to understand if the system is made of 4 or 5 elements, it is not so if we have to discriminate between a system made up of one billion or of one billion and one elements. This assumption seems so obvious and natural that on it, we have implicitly based the very definition of intensive quantity, the concept of thermodynamic limit, and, going to the bare bones, the whole thermodynamics.

On the other hand, physics has also accustomed us to small or huge unexpected results that have shaken from the foundations many of our convictions consolidated over time. In the following, we will take into account a quantum system out of equilibrium and consider a typical measure of its evolution, known as the Loschmidt-echo (LE) [1–4]: we will show how, under some conditions, the Loschmidt-echo (LE), displays two completely different behaviors depending on whether the number of components in the system is even or odd. Quite surprisingly, such a difference, not only does not disappear as the number of elements increases but, to the contrary, it becomes more and more evident moving towards thermodynamically large systems. In other words, the analysis of the LE allows for a discrimination between systems consisting of $N$ and $N+1$ elements even when $N$ is in the order of the Avogadro's number, hence representing a clear violation of what would be expected from a naïve application of the concept of the thermodynamic limit. This result is similar to the even/odd effect in the current–voltage curve observed in superconducting transistors, but it persists to much greater numbers, while

the latter is limited by the capacity of the mesoscopic dot ($10^9$) [5].

One can define the LE as the overlap between a state $|g\rangle$ and its evolution driven by the Hamiltonian that describes the system. We employ a *quantum quench* protocol [4, 6–8], which is one of the most popular ways to drive a system out of equilibrium: namely, we prepare the system in the ground state $|g\rangle$ of an initial Hamiltonian $H_0$ and then we suddenly add a perturbation $\lambda H_p$, where $\lambda$ stands for a tunable amplitude while the eigenvalues of $H_p$ are of the order of unity [2, 9, 10]. Then the initial state is unitarily evolved by the Hamiltonian $H_1 = H_0 + \lambda H_p$ and the LE can be defined as

$$\mathcal{L}(t) = |\langle g|e^{-iH_1 t}|g\rangle|^2. \qquad (1)$$

A deeper insight in the time behavior of the LE can be obtained by expanding the initial state in terms of the eigenstates $|n\rangle$ of the perturbed Hamiltonian $H_1$:

$$\mathcal{L}(t) = \left|\sum_n e^{-iE_n t}|c_n|^2\right|^2, \qquad c_n = \langle n|g\rangle. \qquad (2)$$

In the general (nontrivial) case, the state $|g\rangle$ is not an eigenstate of the Hamiltonian $H_1$ and thus several coefficients $c_n$ assume a non-vanishing value and the time evolution of the LE depends on their relative weights. Roughly speaking, we can arrange the possible behaviors into two large families. The first is made of the cases in which one of the coefficients is much greater, in absolute value, than the sum of all the others. As a consequence, denoting as $|0\rangle$ the eigenstate of $H_1$ for which $c_n$ reaches the maximum, from eq. (2) we recover that the LE will be characterized by oscillations with an average value close to the identity and oscillation amplitudes bounded from above by $(1 - |c_0|^2)|c_0|^2$. On the other hand, if none of the $c_n$ dominates over the others, we obtain an evolution characterized by a more complex pattern with larger oscillation amplitudes.

These two prototypical behaviors for the LE are generally associated with different properties of the physical

systems [12, 13]. For example, the first trend type characterizes systems in which $H_0$ shows an energy gap that separates the ground state from the set of the excited states [3, 11, 14]. In this case, assuming that $\lambda$ is much smaller than the energy gap, the coefficient $\langle g_1|g\rangle$, where $|g_1\rangle$ is the ground state of $H_1$, is expected to be much larger than all the others. On the other side, for systems in which the ground state of $H_0$ is part of a narrow band that, in the thermodynamic limit, tends to a continuous spectrum, the perturbation $\lambda H_p$ may induce a non-negligible population in several low-energy excited states [15] and, hence, the time evolution of the second kind [16].

Typically, these different spectrum properties do not turn into one another by changing the number of elements that make up the system. Indeed, the presence or the absence of a gap in the energy spectrum is related to the different symmetries of the Hamiltonian and, usually, they are not size-dependent. Hence, keeping all other parameters fixed and increasing the number of elements, we expect the same kind of time-evolution, with finite-size effects that reduce with the system size up to some point at which the dependence of the LE on the number of constituents is almost undetectable. To have a LE evolution that changes as the number of elements turns from even to odd and vice-versa, we need to find a system in which also the shape of the energy spectrum is strongly dependent on it.

In the very last years, it was pointed out that such models can be found, among the one-dimensional spin-$1/2$ models with periodic boundary conditions. Namely, they are short-range antiferromagnetic systems in which frustration [17–23] is induced when the number of the elements making up the system is odd, so realizing the so-called frustrated boundary conditions [24–27]. Hence the presence/absence of frustration in the ring geometry (therefore also the term *topological frustration*) is a direct consequence of the fact that the number of the spins is odd/even.

To understand well how such kind of frustration works let us take a step back. In classical antiferromagnetic systems, when the number of the elements is an integer multiple of two, even in the presence of periodic boundary conditions, there is no problems in minimizing the contribution of every single term to the total energy. Therefore, the system will show a ground state manifold made of the two Neel states, well separated from the excited states by an energy gap that does not vanish in the thermodynamic limit. This picture is very resilient and also the introduction of quantum effects does not change it significantly [28, 29]. On the contrary, when the number of elements turns odd, the presence of the periodic boundary conditions makes it impossible to satisfy simultaneously all the local constraints [17, 18]. Such an impossibility induces a frustration that gives rise to the creation of a set of states that are Neel states with a pair of parallelly oriented spin, a so-called domain wall. If the system is invariant under spatial translation, since the defect can be placed equivalently on every lattice site, the ground state manifold of the system becomes highly degenerate, consisting of $2N$ states for a chain made by $N$ sites, and it is separated from the other states by an energy gap that stays finite in the thermodynamic limit. When quantum effects are taken into account, the macroscopic ground-state degeneracy is typically lifted, generating a narrow band of states (which can be interpreted as containing a single excitation with a definite momentum) and thus yielding an energy gap that vanishes in the thermodynamic limit. Hence, as a result, the energy spectrum of such models depends dramatically on whether the number of elements is even or odd.

However, this property alone is not sufficient to ensure a dependence of the dynamics of the LE on the size of the system like the one we are looking for. The perturbation that acts on the initial Hamiltonian must also be chosen carefully. On the one hand, as the states in the lowest energy band of the frustrated system are identified by different quantum numbers (namely, their momenta), the perturbation should break the symmetry these numbers reflect, to ensure that the eigenstates of the perturbed Hamiltonian can have a finite overlap in the whole band (otherwise, we expect that the initial state would overlap only with states carrying the same quantum number). On the other hand, if the unfrustrated system is in a symmetry broken phase with an (asymptotically) degenerate ground state manifold, we want the perturbation to preserve the ground state choice, so that in the evolution the overlap between other ground state vectors remain suppressed.

**Results:** To clarify these arguments and to provide a specific example, let us discuss a paradigmatic model i.e. the antiferromagnetic Ising chain in a transverse magnetic field with periodic boundary conditions [30–32]. Such well-known model is described by the following Hamiltonian

$$H_0 = \sum_{j=1}^{N} \left( \sigma_j^x \sigma_{j+1}^x + h \sigma_j^z \right). \tag{3}$$

Here $\sigma_j^\alpha$ with $\alpha = x, y, z$ stands for the Pauli operators defined on the $j$-th lattice site, $h$ is the relative weight of the local transverse field, $N$ is the length of the ring and periodic boundary conditions imply that $\sigma_{N+j}^\alpha = \sigma_j^\alpha$. As we can see from eq. (3), the system holds the parity symmetry with respect to the $z$-spin direction, since $[H_0, \Pi^z] = 0$ where $\Pi^z = \bigotimes_{i=1}^{N} \sigma_i^z$. This means that the eigenstates of $H_0$ can be arranged in two sectors, corresponding to two different eigenvalues of $\Pi^z$. Moreover, the model in eq. (3) also holds an invariance under spatial translation which implies that there exists a complete set of eigenstates of $H_0$ made of states with definite lattice momentum [33].

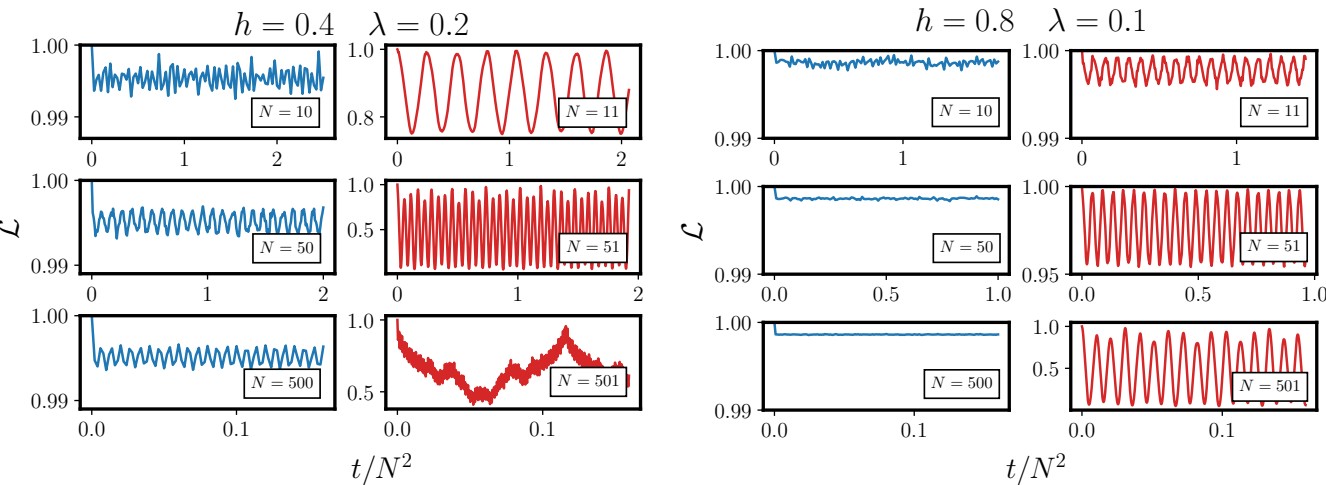

FIG. 1: (Color online) Loschmidt echo comparison between frustrated and unfrustrated chains of similar length $N$, fixing the magnetic field and the perturbation parameter respectively to $h = 0.4$, $\lambda = 0.2$ (left plot) and $h = 0.8$, $\lambda = 0.1$ (right plot). The time is rescaled for a better comparison. For even $N$ (unfrustrated systems), due to the negligible hybridization with the first excited states, the LE presents small oscillations around a value near one (left columns). For odd $N$ instead the higher number of hybridized states results in a strong sensitivity of the LE oscillations to the system parameters.

In the range $0 < h < 1$, for $N = 2M$ the system shows two nearly degenerate lowest energy states with opposite parity and an energy difference closing exponentially with the system size [30, 34] while all other states remain separated by a finite energy gap. When $N = 2M + 1$, topological frustration sets in and the unique ground states becomes part of a band in which states of different parities alternate. In this case the gaps between the lowest energy states close algebraically as $1/N^2$ [25, 26, 35–39].

A simple perturbation that satisfies the criteria we discussed above is $H_p = \lambda \sigma_N^z$, since it breaks the translational invariance which classifies the eigenstates of $H_0$, while preserving the parity symmetry. Thus, we have

$$H_1 = H_0 + \lambda \sigma_N^z, \qquad (4)$$

and we assume that $\lambda \ll 1$.

Since $H_1$ is not invariant under spatial translation, we cannot diagonalize it analytically as is possible for $H_0$, by exploiting the usual approach based on the Jordan-Wigner transformation followed by a Bogoliouv rotation [30]. Nevertheless, we can resort to the diagonalization procedure reported in [28], which allows us to diagonalize numerically the Hamiltonians eq.s (3), (4) in an efficient way [40] and thus to calculate the LE (see the Methods section for the details). The results obtained are summarized in Fig. 1, where several behaviors of the LE for even $N$ and odd $N + 1$ sizes are compared.

The results fit well in the qualitative picture we have discussed in the first part. When $N$ is even and hence the system is not frustrated the LE presents small noisy oscillations around a value close to unity, see Fig. 1. The average value is almost independent from the parameters, while oscillations reduce as the system size increases. This behavior reflects the fact that, $\lambda$ being small, the initial state shares a significant overlap only with one of the lowest eigenstates of $H_1$ and the contributions from all other states above the gap produce fast oscillations that average out in the long time limit.

For the frustrated case $N = 2M + 1$ instead, the picture is completely different. Here, because of the closing of the gap, the same perturbation hybridizes several states, which thus contribute to the evolution of the LE. Finite-size effects become important, since the density of states changes with the chain length approaching its asymptotic value and thus changes the number of states which get hybridized. These considerations imply a strong sensibility in the LE's oscillation frequency and amplitude to all the parameters in the setting.

The results presented in Fig. 1 make it clear that the behaviors of the LE for even and odd $N$ are completely different. To go beyond this qualitative assessment, we can make a quantitative comparison of the difference between these two behaviors, by considering the time averaged value of the LE $\bar{\mathcal{L}}$ over a long period of time, which ideally tends to be infinite. Such analysis, whose results can be appreciated in the left panel of Fig. 2, clearly shows that for the unfrustrated case (blue circles) the time average is almost independent from the size of the ring, while for the frustrated one (red squares) there is a significant dependence on the ring's size with an asymp-

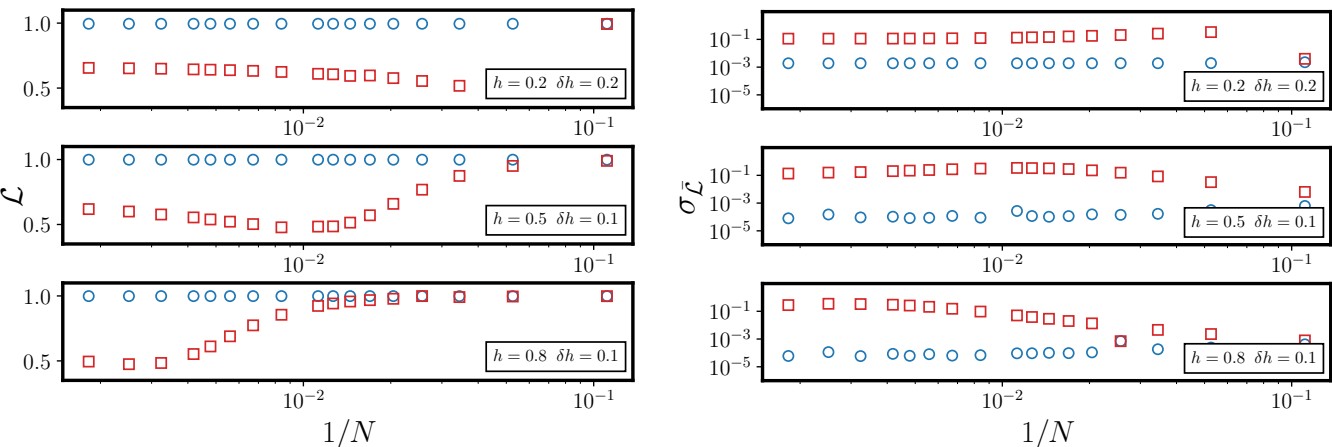

FIG. 2: (Color online) Comparison between the result for frustrated (red squares) and unfrustrated (blue circles) chains of the time-average (left panel) and of the standard deviations (right panel) of the LE for several sets of parameters as a function of the inverse system length. Differently from the frustrated case, the unfrustrated time average is mostly size independent. The standard deviation deviation for the frustrated case is always larger, even a few orders of magnitude, than the one of the unfrustrated case.

totic value in the thermodynamic limit which differs from the even chain length case. The similarity between the frustrated and unfrustrated values for small systems can be easily understood considering that for small $N$ the gap between the ground state and the other states in the lowest energy band in the frustrated models can be bigger than the perturbation amplitude hence giving life to an "unfrustrated-like" behavior for the LE.

As we wrote above, since $H_p$ breaks the spatial invariance, it is impossible to obtain an exact expression for the LE. For the unfrustrated case, it is possible to develop a cumulant expansion [3] which provides the correct evolution of the LE, but its reliability hinges on a clear separation of scales between the strength of the perturbation and the energy gap. When $N$ is odd, the gap closes and for sufficiently large system size this approach fails. Nonetheless, to gain some insight into the LE when the system is frustrated, we can resort to a perturbation theory around the classical point ($h = 0$) [33, 41, 42] and derive an analytic expression which can be compared to our numerical results. Within this approach, we first compute the initial (ground) state of $H_1$ considering, in the beginning, $\lambda \sigma_N^z$ as the perturbation to the Hamiltonian at the classical point ($h = 0$), and then bringing back the term $h \sum_j \sigma_j^z$ as a second-order perturbation term. By construction, this approach is justified for $0 < h \ll \lambda \ll 1$. The effect of the local term $\lambda \sigma_N^z$ is to split the initial $2N$ degenerate states into three groups. In particular, the ground space becomes two-fold degenerate, separated by an energy gap of order $\lambda$ from $2N - 4$ degenerate states, on top of which, separated by a gap of the same value, there are two other degenerate states. The second perturbation term $h \sum_j \sigma_j^z$ does not act significantly on the two two-dimensional manifolds but removes the macroscopic degeneracy, creating an intermediate band of $2N - 4$ states.

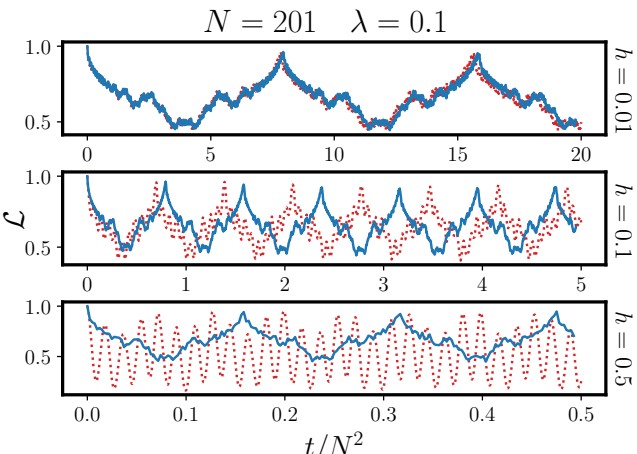

FIG. 3: (Color online) Loschmidt echo's comparison between the numerics (dotted red line) and the analytic expression eq. (5 (blue line) for a spin chain of length $N = 201$ and for $\lambda = 0.1$. The time is rescaled for a better comparison. The results are in agreement for $h = 0.01$, that corresponds to the limit $0 < h \ll \lambda \ll 1$ (upper panel, the curves are mostly superimposed). We also find similar results when $h$ and $\lambda$ are comparable, as shown in the middle panel for the case $\lambda = h = 0.1$. Finally in the lower panel it is shown the failure of the approximation for $h = 0.5$ when the value of the magnetic field is beyond the assumed range of validity.

Exploiting this perturbative analysis (see the Method section for details), we obtain for the LE

$$\mathcal{L}(t) = \left| \frac{2}{N(N-1)} \sum_{k=1}^{(N-1)/2} \tan^2 \left[ \frac{(2k-1)\pi}{2(N-1)} \right] \exp \left\{ -i2ht \cos \left[ \frac{(2k-1)\pi}{N-1} \right] \right\} + \frac{2}{N} \exp \left[ it(\lambda+h) \right] \right|^2. \qquad (5)$$

In Fig. 3 we compare the analytical results eq. (5) with the numerical data and we find a substantial agreement between the two in the region, especially for $h \ll \lambda$, see the upper panel. It is also worth noting that the two methods give similar results even when $h$ and $\lambda$ are comparable (middle panel of Fig. 3). The main difference between the two behaviors is, apparently, only a rescaling of the oscillation frequency that seems to be underestimated in the perturbative approach.

In the thermodynamic limit the term proportional to $2/N$ in eq. (5) can be neglected and the expression of the LE can be approximated as: $\mathcal{L}(t) \simeq \mathcal{F}\left(\frac{2ht}{N^2}\right)$ where the function $\mathcal{F}(x)$ is given by

$$\mathcal{F}(x) = \lim_{M \to \infty} \left| \frac{1}{2M^2} \sum_{k=1}^{M} \tan^2 \left[ \frac{(2k-1)\pi}{4M} \right] \times \right. \qquad (6)$$

$$\left. \times \exp \left\{ -ix(2M+1)^2 \cos \left[ \frac{(2k-1)\pi}{2M} \right] \right\} \right|^2$$

The function in eq. (6) is somewhat reminiscent of the Weierstrass function [43] and indeed it displays a continuous, but nowhere differentiable behavior. While it is remarkable its emergence in such a simple context, we remark that such fractal curve [44, 45] was already observed in LE evolution [16]. Moreover, it is worth noting that the expression in eq. (5) is very far away from the one obtained by Silva [3] for systems with an even number of elements where, in the thermodynamic limit, the oscillations are suppressed and the LE assumes the form of a straight line.

In conclusion, we analyzed the behavior of the LE in short-range antiferromagnetic one-dimensional spin systems with periodic boundary conditions in the presence of a perturbation that violates translational invariance, but leaves unaffected the parity, namely a local magnetic field. Under these conditions, the LE shows an anomalous dependence on the number of elements in the system. When this number is even, LE shows small random oscillations around a value very close to unity that is almost independent from the system size, and the amplitude of these oscillations tend to decrease with the size increasing until it disappears in the thermodynamic limit. On the contrary, in the presence of a ring made out of an odd number of sites, the oscillations are large and do not disappear in the thermodynamic limit while the average value is strongly dependent on the system size. The presence of two different behaviors can be traced back to the different energy spectrum that arises from the presence, in the case of an odd number of elements, of a topological frustration that leads to a closure of the energy gap, which is instead finite in these systems when $N$ is even. These general results have been tested in a paradigmatic model, the Ising model in the transverse field, using both exact diagonalization methods and perturbation theory.

This peculiar LE trend represents an exception to the behavior of intensive quantities in the thermodynamic limit. Within this limit, as it is well known, intensive quantities tend to assume a constant value in which the dependence on the system size disappears. On the contrary, in the case just analyzed this does not happen as the LE flips continuously between the two trends depending on whether the number of elements is even or odd. This behavior is reminiscent of the current-voltage curve in superconducting transistors [5], which show periodic $2e$ modulations depending on the parity of the total number of electrons in the superconducting island, but it is not limited to mesoscopic systems and extends to arbitrary large sizes. This result is even more relevant when we take into account that LE plays a fundamental role in several problems of current interest in quantum thermodynamics such as quantum work statistics [3, 46] and information scrambling [46, 47]. Indeed, a detailed analysis of the implication of this work in these applications and additional quantitative characterizations of the frustrated LE will be the subject of future works. Furthermore, the LE can be easily observed experimentally by looking at the decoherence of a two-level system interacting with the spin system [11].

On the other hand, this particular LE pattern can be seen as a further surprise provided by one-dimensional systems with topological frustration. Despite their simplicity, they present several peculiar aspects such as incommensurate magnetic patterns [33, 41], the appearance of phase transitions not present with boundary conditions that do not force the presence of frustration [33, 42], etc. Until now, the analysis had focused on the static aspects induced by topological frustration. However, our work also emphasizes that this violent change between systems with different spectral properties can greatly influence the dynamics and open the door to possible applications of such models in the perspective of quantum computing [24] as well as in quantum thermodynamics.

## ACKNOWLEDGMENTS

SMG, FF, and GT acknowledge support from the QuantiXLie Center of Excellence, a project co–financed by the Croatian Government and European Union through the European Regional Development Fund – the Competitiveness and Cohesion (Grant KK.01.1.1.01.0004). VM, DK, SMG, and FF also acknowledge support from the Croatian Science Foundation (HrZZ) Projects No. IP–2016–6–3347. SMG and FF also acknowledge support from the Croatian Science Foundation (HrZZ) Projects No. IP–2019–4–3321.

## METHODS

*Loschmidt echo.* Let us provide a detailed description of the method exploited to obtain the data on the Ising model plotted in the paper. Our starting point is to observe that, for spin systems that can be mapped to free-fermionic models, eq. (1) can be rewritten in the following form [11, 14]:

$$\mathcal{L}(t) = |\det(1 - \mathbf{r} + \mathbf{r}e^{-i\mathbf{C}t})|. \quad (7)$$

Here

$$\mathbf{\Delta}^\dagger = (c_1^\dagger, \ldots, c_N^\dagger, c_1, \ldots, c_N), \quad (8)$$

describes the fermionic operators, $\mathbf{C}$ is the matrix coefficient of the Hamiltonian $H_1$ in the fermionic language, i.e.

$$H_1 = \frac{1}{2}\mathbf{\Delta}^\dagger \mathbf{C} \mathbf{\Delta}, \quad (9)$$

and $\mathbf{r} = \langle g|\Delta_i^\dagger \Delta_j|g\rangle$ is the two-point fermionic correlation matrix in the initial state. The hermiticity requirement for the Hamiltonian fixes the matrix $\mathbf{C}$ to be of the block-form

$$\mathbf{C} = \begin{pmatrix} \mathbf{S} & \mathbf{T} \\ -\mathbf{T} & -\mathbf{S} \end{pmatrix}, \quad (10)$$

where $\mathbf{S}$ is a symmetric and $\mathbf{T}$ an antisymmetric matrix.

It is useful to rewrite the $\mathbf{r}$ matrix in terms of the correlation functions of the Majorana operators. Following [28] we define:

$$A_i = c_i^\dagger + c_i \quad (11)$$
$$B_i = i(c_i^\dagger - c_i). \quad (12)$$

Exploiting eq. (11) and eq. (12) and the fact that, since $|g\rangle$ is the ground state of $H_0$, $\langle g|A_iA_j|g\rangle = \langle g|B_iB_j|g\rangle = \delta_{ij}$ it is straightforward to obtain:

$$\mathbf{r} = \frac{1}{4}\begin{pmatrix} 2\mathbf{I} + \mathbf{G} + \mathbf{G}^\intercal & \mathbf{G} - \mathbf{G}^\intercal \\ -\mathbf{G} + \mathbf{G}^\intercal & 2\mathbf{I} - \mathbf{G} - \mathbf{G}^\intercal \end{pmatrix}. \quad (13)$$

with $G_{ij} = -i\langle g|B_iA_j|g\rangle$.

Therefore, to calculate the LE it remains to evaluate the correlation matrix $G$ on the ground state of the unperturbed Hamiltonian $H_0$ and the matrix $C$ linked to $H_1$. Both can be determined following the same approach. Exploiting the Jordan-Wigner transformation

$$c_j = \left(\bigotimes_{l=1}^{j-1}\sigma_l^z\right)\frac{\sigma_j^x + i\sigma_j^y}{2}, \quad c_j^\dagger = \left(\bigotimes_{l=1}^{j-1}\sigma_l^z\right)\frac{\sigma_j^x - i\sigma_j^y}{2}, \quad (14)$$

we map the spin system to a quadratic fermionic one. In fact, due to non-locality of the Jordan-Wigner transformation the Hamiltonians eq. (3) and eq. 4 cannot be written as a quadratic form eq. (9). However, they commute with the parity operator $\Pi^z = \bigotimes_{i=1}^N \sigma_k^z$ and it is possible to separate them into two parity sectors, corresponding to the eigenvalues $\Pi^z = \pm 1$, so that in each sector they are a quadratic fermionic form. In the following, we can restrict ourselves to the Hamiltonians $H_0$ and $H_1$ only in the odd sector ($\Pi^z = -1$) since the ground state of the quantum Ising model eq. (3) with frustrated boundary conditions and $h > 0$ belongs to it [26, 35]. There, they can be written in the form of eq. (9), up to an additive constant. In particular, the matrix $\mathbf{C}$ for $H_1$ in the odd sector, present in eq. (7), can be obtained easily by inspection.

The matrix $\mathbf{G}$ can be found easily from the exact solution of the quantum Ising chain with frustrated boundary conditions [26, 35]. However, for a more efficient numerical implementation we follow the approach from ref. [14, 28], where we write $H_0$ in the odd sector in the form of eq. (9) and where

$$G_{ij} = -(\mathbf{\Psi}^\intercal \mathbf{\Phi})_{ij}, \quad (15)$$

with the matrices $\mathbf{\Phi}$ and $\mathbf{\Psi}$ being formed by the corresponding vectors given by the solution of the following coupled equations:

$$\Phi_k(\mathbf{S} - \mathbf{T}) = \Lambda_k\Psi_k, \quad (16)$$
$$\Psi_k(\mathbf{S} + \mathbf{T}) = \Lambda_k\Phi_k. \quad (17)$$

This problem can be easily solved considering the following eigenvalue problems:

$$\Phi_k(\mathbf{S} - \mathbf{T})(\mathbf{S} + \mathbf{T}) = \Lambda_k^2\Phi_k, \quad (18)$$
$$\Psi_k(\mathbf{S} + \mathbf{T})(\mathbf{S} - \mathbf{T}) = \Lambda_k^2\Psi_k. \quad (19)$$

Here the eigenvalues give us the free-fermionic energies $\Lambda_k$. The sign of a particular energy is a matter of choice. Transforming $\Lambda_k$ to $-\Lambda_k$ corresponds simply to switching the creation and the annihilation operator, and to transforming $\Phi_k$ ($\Psi_k$) into $-\Phi_k$ ($-\Psi_k$) in eq.s (16) and (17). It is important to note that the parity requirements do not allow for the ground state of $H_0$ to be the vacuum state for free fermions with positive energy [26, 35]. Thus,

assuming the eigenvalues of the matrix appearing on the l.h.s. of eqs. (18), (19) are labeled in ascending order, the ground state corresponds to the vacuum state of fermions with $\Lambda_1 < 0$ and the remaining energies $\Lambda_k$ positive.

*Perturbation Theory near the classical point* Let us now turn to provide some more details on the perturbative approach to the LE near the classical point in the presence of topological frustration. The first step consists of finding the ground state of the Hamiltonian $H_0$ in eq. (3), treating the term $h \sum_j \sigma_j^z$ as a perturbation. It is known that, at the classical point, in the presence of an odd number of spins the interplay between periodic boundary conditions and antiferromagnetic interactions gives rise to a $2N$-fold degenerate ground state manifold. Such a space is spanned by the kink states $|j\rangle$ and $\Pi^z |j\rangle$, $j = 1, 2, \ldots N$ with energy $-(N-2)$, that have one ferromagnetic bond $\sigma_j^x = \sigma_{j+1}^x = \pm 1$ respectively, the others being antiferromagnetic ($\sigma_k^x = -\sigma_{k+1}^x$ for $k \neq j$). The excited states outside this manifold are separated from the ground space by an energy gap of order unity so that we can neglect them in a perturbative approach. By considering the magnetic field the $2N$-fold degeneracy is removed and a narrow-band of states is created, with a gap that separates the ground state from the other elements of the band closing as $1/N^2$ (see Ref. [27, 33]). To the lowest order in perturbation theory in $h$ we found for the initial state appearing in eq. (1), that is for the ground state of the unperturbed system, the expression:

$$|g\rangle = \frac{1}{\sqrt{N}} \sum_{j=1}^{N} \frac{1 - \Pi^z}{\sqrt{2}} |j\rangle. \qquad (20)$$

The next step is to find the lowest energy states of the Hamiltonian $H_1$ in eq. (4) through a perturbation theory both for $h > 0$ and $\lambda > 0$. Since we first apply the perturbation theory in $\lambda$ while we consider $h$ as a second-order perturbation, we are assuming that $h \ll \lambda \ll 1$. Also in this case we start again from the $2N$ degenerate ground space formed by the kink states and we treat the term $\lambda \sigma_N^z$ as a perturbation. Again we find that the degeneracy is removed and, at this point, the system shows a two-fold degenerate with eigenvectors given by:

$$|\psi_\pm\rangle = \frac{1 \pm \Pi^z}{2} (|N-1\rangle \mp |N\rangle), \qquad (21)$$

separated by an energy gap equal to $\lambda$ from $2N-4$ degenerate kink states. Above this macroscopically degenered manifold, separated by a gap $\lambda$ there are other two states:

$$|\phi_\pm\rangle = \frac{1 \pm \Pi^z}{2} (|N-1\rangle \pm |N\rangle) \qquad (22)$$

We now consider the second-order perturbation $h \sum_j \sigma_j^z$. Its effect on the $|\psi_\pm\rangle$ and $|\phi_\pm\rangle$ states is only a shift in the energy respectively of $\mp h$. Furthermore it creates a

band of states from the kink ones given by:

$$|\xi_\pm, m\rangle = \frac{1 \pm \Pi^z}{\sqrt{N-1}} \sum_{j=1}^{N-2} (-1)^j \sin\left(\frac{m\pi}{N-1}j\right) |j\rangle, \quad (23)$$

with $m = 1, 2, \ldots, N-2$. The energies of the discussed eigenstates are given by

$$E(\psi_\pm) = -(N-2) - (\lambda + h), \qquad (24)$$
$$E(\phi_\pm) = -(N-2) + \lambda + h, \qquad (25)$$
$$E(\xi_\pm, m) = -(N-2) \mp 2h \cos\left(\frac{m\pi}{N-1}\right). \qquad (26)$$

The calculation of the Loschmidt echo is now straightforward. From the definition eq. (1), expressing the initial state eq. (20) in terms of the eigenstates of the perturbed model eq.s (21), (22), and (23) and applying the evolution operator $e^{-iH_1 t}$ we finally obtain the expression in eq. (5).

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
