# Peer review of "An odd thermodynamic limit for the Loschmidt echo"

_SciPost Physics_

## Round 1 · Referee Report · Anonymous (Referee 1) · 2021-8-12

Strengths

see report

Weaknesses

see report

Report

The authors investigate even/odd effects in the quench dynamics of large periodic spin chains using the Loschmidt echo as a probe. They demonstrate that the dynamical response for systems showing what they call 'topological frustration' can be markedly different depending on whether the number of sites in the chain is even or odd. The example studied is the transverse Ising chain where the quench consists in modulating the strength of the local transverse field on a single site, thus breaking translational invariance. The results are explained both in a qualitative, physical picture and also by performing a perturbative expansion.

All the obtained results appear solid and the idea of testing such even-odd effects using a quantum quench and the Loschmidt echo is of interest. It appears fair to say that this does provide a new synergetic link to the field of quantum dynamics. The paper in its current version, in my view, does however not provide an abstract and introduction which puts the problem into the proper context nor does it seem to provide citations in a way which is representative and complete, see below. However, it appears possible to remedy these issues and an amended manuscript might then be reconsidered for publication in Scipost Physics.

What I find problematic is the abstract and the introduction. First, it appears to me that the authors are trying unnecessarily hard to sell their results. The paper discusses an interesting phenomena but no foundations will be shaken here as the introduction appears to imply. Second, the discussion of intensive and extensive thermodynamic quantities and that those cannot distinguish between "one billion" and "one billion and one elements" appears to me to be missing the point. Regular intensive and extensive thermodynamic quantities do characterize the macrostate of a system. I.e., these are local probes which cannot distinguish between individual microstates. In contrast, the Loschmidt echo studied here is a probe of the microstate itself. For individual quantum states we are actually quite used to the notion that they can be very sensitive to adding or removing a single particle or a single site. While these are not to be understood as exact analogies to the effect studied here, we do understand, for example, that the removal of a single particle in the X-ray edge problem leads to the Anderson orthogonality catastrophe or that the addition or removal of a single site can change the number of edge modes in a topological insulator and thus drastically alter their ground-state entanglement properties.

I have two recommendations:

1) To rewrite the abstract and the introduction for the reasons listed above.

2) There is a large literature on the Loschmidt echo for many-body systems. At least some of this literature seems to be of relevance to put the results better into context. Just to give one example: here the authors consider a local quench in the transverse Ising chain. Global quenches have been considered as well. Is it important that the quench is local? One could also break translational invariance by having a staggered, modulated, or small random field component on all sites.

Requested changes

see report

  • validity: high
  • significance: good
  • originality: high
  • clarity: low
  • formatting: excellent
  • grammar: good

Author:  Fabio Franchini  on 2021-10-06  [id 1817]

(in reply to Report 1 on 2021-08-12)
Category:
reply to objection

We thank the referee for their reading and assessment of our manuscript. We apologize because indeed our excitement prevent us from placing our work in the right context.
We appreciate and we accepted the suggestions from the referee and rewrote the paper (and in particular the introduction) to lower our tone and better frame our results within the existing literature.
We hope that the new version of our paper will satisfy the referee.

---

## Round 1 · Referee Report · Anonymous (Referee 2) · 2021-8-24

Strengths

See Report

Weaknesses

See Report

Report

The results presented in the manuscript are interesting but not surprising. The crux of the paper is the following:

  1. Within periodic boundary conditions, whether the antiferromagnetic spin chain is frustrated or not depends on the number of spins (odd/even) in the chain.

  2. The energy spectrum of the system is determined by the presence or absence of frustration in the system.

  3. The nature of the energy spectrum in turn determines the evolution of the Loschmidt echo (LE) following a local quench which preserves parity but breaks the translational invariance.

Although the results are correct and thorough. In addition to the fact that the results have been demonstrated for a special situation (local quench and Ising system), I do not see the significance or general applicability of the results itself. In particular, the authors argue the importance by highlighting the applicability of the Loschmidt echo in a number of other areas of research, but it is not apparent as to how the sensitivity of the LE to odd/even parity can have any significant impact.

The authors may consider discussing if similar results can be expected for intrinsically frustrated systems (such as the ANNNI model). This would make the results more robust.

In short, I do not recommend the publication of the manuscript in the present form. The manuscript needs to be rewritten addressing the issues mentioned above. Specifically, the impact of the results should be illustrated at least through an example.

Requested changes

The impact of the results should be illustrated at least through an example.

The authors may consider discussing if similar results can be expected for intrinsically frustrated systems (such as the ANNNI model).

  • validity: good
  • significance: good
  • originality: good
  • clarity: high
  • formatting: good
  • grammar: good

Author:  Fabio Franchini  on 2021-10-06  [id 1816]

(in reply to Report 2 on 2021-08-24)
Category:
answer to question
reply to objection

We thank the referee for their reading and assessment of our manuscript. We apologize for not being sufficiently clear in the first version regarding the meaning and significance of our results. In the new version we rewrote the introduction completely and modified the body to amend our original shortcomings. However, let us stress here what we think are the main messages we would like to convey: - It has been recognized long ago that, with AFM interactions, odd (even) chain lengths with periodic boundary conditions lead to the appearance (disappearance) of geometrical frustration, which, in turns, changes the low energy spectrum of the model. Although this effect is generic for many AFM spin chain, it has been largely deemed inconsequential; - It has also been discussed in the recent literature how the Loschmidt Echo for small quantum quenches displays different qualitative signatures depending on the low energy spectrum, which has been mostly used as a way to detect quantum phase transitions; - In our work we observe that, for the same parameter choice, different qualitative behaviors of the LE can be distinguished as due solely to the parity of the chain length, thus finally providing an (experimentally) observable way to detect the presence of geometrical (topological) frustration and settling a decades long dispute on its direct observability; - As we argue in the second section of the revised manuscript, the protocol we employ is generic and driven by the goal we wish to accomplish. The same applies to the model we choose as an example (the Ising chain in a transverse field). Any model with purely topological frustration and any quench that breaks translational invariance would yield the same results. Instead, systems with an extensive amount of frustration (such as the ANNNI model in part of its phase diagram) have a gapless spectrum regardless of boundary conditions and thus are not sensitive to the parity of the chain length; - The LE is a proxy to detect several interesting properties of systems out of equilibrium and is thus a quantity of primary importance for quantum energetics and quantum thermodynamics. Our results mean that for odd chain length the system displays larger fluctuations which can be harvested in several protocols. We have already determined settings in which a frustrated system outperforms its even chain length counterpart and we will report on these studies soon. We hope that these clarifications, as well as the new version of our paper, will convince the referee to change their opinion and decide favorably toward the acceptance of our work.

---

## Editorial Decision

resubmitted